# Exploring the Steps of Infrared (IR) Spectral Analysis: Pre-Processing, (Classical) Data Modelling, and Deep Learning

**DOI:** 10.3390/molecules28196886

**Published:** 2023-09-30

**Authors:** Azadeh Mokari, Shuxia Guo, Thomas Bocklitz

**Affiliations:** 1Leibniz Institute of Photonic Technology, Member of Research Alliance “Leibniz Health Technologies”, 07745 Jena, Germanyshuxia.guo@uni-jena.de (S.G.); 2Institute of Physical Chemistry, Friedrich Schiller University Jena, 07743 Jena, Germany; 3Institute of Computer Science, Faculty of Mathematics, Physics & Computer Science, University Bayreuth, Universitaet sstraße 30, 95447 Bayreuth, Germany

**Keywords:** infrared spectroscopy, pre-processing techniques, data modeling, deep learning, machine learning, analytical technologies, baseline, noise, artificial intelligence

## Abstract

Infrared (IR) spectroscopy has greatly improved the ability to study biomedical samples because IR spectroscopy measures how molecules interact with infrared light, providing a measurement of the vibrational states of the molecules. Therefore, the resulting IR spectrum provides a unique vibrational fingerprint of the sample. This characteristic makes IR spectroscopy an invaluable and versatile technology for detecting a wide variety of chemicals and is widely used in biological, chemical, and medical scenarios. These include, but are not limited to, micro-organism identification, clinical diagnosis, and explosive detection. However, IR spectroscopy is susceptible to various interfering factors such as scattering, reflection, and interference, which manifest themselves as baseline, band distortion, and intensity changes in the measured IR spectra. Combined with the absorption information of the molecules of interest, these interferences prevent direct data interpretation based on the Beer–Lambert law. Instead, more advanced data analysis approaches, particularly artificial intelligence (AI)-based algorithms, are required to remove the interfering contributions and, more importantly, to translate the spectral signals into high-level biological/chemical information. This leads to the tasks of spectral pre-processing and data modeling, the main topics of this review. In particular, we will discuss recent developments in both tasks from the perspectives of classical machine learning and deep learning.

## 1. Introduction

IR technology measures the interaction between infrared light and the sample, for example, the molecules that make up the sample. A molecule can absorb infrared radiation via three different mechanisms, all of which result in an increase in energy within the molecule proportional to the amount of light absorbed. The first mechanism involves a rotational transition where the molecule is activated to a higher rotational energy level. The second mechanism occurs when the molecule reaches a higher vibrational level after absorption. The third pathway involves an electronic transition, where an electron in the molecule moves to a higher electronic state. Conventionally, IR spectroscopy usually refers to the second mechanism, where the vibrational characteristics of molecules are detected and used for molecular identification. The IR region ranges from 150 to 0.12 KJ mol^−1^ (12,500–10 cm^−1^) [1], and is typically divided into three regions: the near-IR region (12,500–4000 cm^−1^), the mid-IR region (4000–200 cm^−1^), and the far-IR region (200–10 cm^−1^) [1,2,3], each detecting a different molecular vibration type. Typically, an IR spectrum is visualized as absorbance of IR radiation by the sample versus a frequency, wavenumber, or wavelength. Usually, the absorbance is plotted against wavenumbers (in inverse centimeters) on the x-axis (Figure 1). 

Different vibrational modes can be investigated with mid-IR spectroscopy, as shown in Figure 2, and different instruments for such measurement exist. Examples of such devices are dispersive infrared spectrometers (DS) or Fourier-transform infrared spectrometers (FTIR). 

In both types of spectrometers, the energy changes before and after the interaction of the infrared radiation with a sample are measured by recording IR spectroscopic data at different wavenumbers/frequencies. Dispersive infrared spectrometers use a dispersive element like a prism or a grating to split the infrared radiation into different wavelengths and measure the intensity change at each wavelength separately. Fourier-transform infrared spectrometers use an interferometer to generate an interferogram, which is then transformed into a spectrum using mathematical algorithms. Each type of infrared spectrometer has advantages and disadvantages, and the choice depends on the specific application and budget. Dispersive infrared spectrometers are typically less expensive and have better resolution, while Fourier-transform infrared spectrometers are more sensitive and faster. A recently reported technology, optical-photothermal infrared (O-PTIR), enables high-resolution imaging with the help of photothermal effect. Therein, a pulsed laser is used to irradiate a sample, producing a thermal expansion and a transient infrared signal. The infrared signal is detected and analyzed to generate an image of the sample’s chemical composition via submicron spatial resolution. The high spatial resolution (approximately 450 nm), no need for sample preparation, and comparable spectral results to traditional FTIR make it highly applicable for diverse samples [6]. 

IR spectroscopy has proven to be a versatile technique due to its label-free, non-invasive, non-destructive properties and its sensitivity to molecular changes [7,8]. It has found successful applications across large variations of biological samples ranging from intact bacteria and eukaryotic cells to body fluids and tissues. Notably, its potential for cancer diagnosis has gained more and more attention, for instance, in breast tumor diagnosis conducted by Haka et al. [9]. Proof of principle investigations, as reviewed by Kondepati et al. [10], have demonstrated the capacity of IR spectroscopy to identify malignant disease states using serum samples. In fact, IR spectroscopy via collecting the biofluid spectral fingerprint has provided a simple, rapid, and label-free manner of detecting cancers and other diseases [11,12,13,14]. However, due to the heterogeneous nature of biological systems, IR spectra are the summation of the contribution of various biomolecules, including proteins, lipids, nucleic acids, carbohydrates, etc. Moreover, an IR measurement not only detects the absorption of the sample but also other physical effects, including scattering, reflection, and interference, that contribute to the signal. These processes produce baseline, band distortion and intensity changes in the measured IR spectra. These facts make the data interpretation extremely challenging. Sophisticated data mining techniques are indispensable to apply IR spectroscopy in practice [15]. Among those, data pre-treatment is necessary to remove unwanted contributions and hence improve the data quality. Data modeling is utilized to relate and translate spectral signals into high-level biological information that helps to draw meaningful conclusions. To make a clear roadmap of the IR spectral analysis, we discuss the concepts of the techniques for pre-processing and data modeling, including approaches of both classical machine learning and deep learning.

## 2. Pre-Processing

Interpretation of IR spectra is often challenging due to contaminations from other effects, as mentioned above. Various pre-processing steps are typically applied to the raw IR data prior to multivariate analysis to overcome these obstacles. These pre-processing steps aim to improve the signal-to-noise ratio and eliminate unwanted signals such as fluorescence, Mie scattering, detector noise, calibration errors, cosmic rays, and laser power variations [16]. The most common pre-processing steps in IR spectroscopy are exclusion, filtering or derivation, baseline correction, and normalization, as shown in Figure 3. Although a large number of approaches have been developed, it can be difficult to determine which pre-processing approach is most appropriate for a given data set. Furthermore, in cases where a single technique is insufficient, it may not be clear which combination of pre-processing steps and in what order will achieve optimal results.

### 2.1. Exclusion (Cleaning)

Exclusion or cleaning is one of the pre-processing methods used in infrared spectroscopy. Its primary function is to ensure the data’s integrity and quality by removing anomalies. In the context of IR spectroscopy, the data can sometimes contain irregularities or noise due to various reasons like instrumental errors, environmental factors, or sample inconsistencies. These irregularities, often referred to as spectral outliers, can skew results and lead to inaccurate interpretations. Exclusion works by detecting and removing these outliers from the dataset [16]. Various methods can be used to assess spectral quality, aiming to identify these outliers before further data processing or analysis occurs. One well-established method to address this issue involves examining the signal-to-noise ratio (SNR). By setting an appropriate threshold on the SNR, researchers can effectively detect and subsequently manage anomalous spectra, ensuring that only high-quality, reliable data are used in subsequent analyses. It is possible to obtain this in a new data space obtained with a feature extraction method such as PCA [17]. If a spectrum stands out significantly from others in the dataset concerning certain variables in the compressed domain—for example, if its intensity is either too low or too high compared to a set threshold for that variable—then it is likely an outlier. Typically, this compressed domain is determined using factor analysis methods like principal component analysis (PCA), which is further discussed in detail in subsequent sections of this review.

### 2.2. Filtering

Filtering, within the context of infrared spectroscopy, is a suite of techniques designed to refine the IR spectral signal by eliminating undesirable components [18]. The techniques range from derivative methods to various computational algorithms, each designed to address a specific contamination in the raw data. The choice of filtering method depends on the specific challenges and artifacts contained in the raw data, be it background fluctuations, noise, or other anomalies. 

#### 2.2.1. Derivative Filters

One prominent filtering method is the calculation of derivatives. Derivative filtering acts as a high-pass filter, emphasizing rapid changes in the signal and thus reducing slower intensity fluctuations. Therefore, contributions from the baseline can be effectively reduced by calculating the first or second derivatives of IR spectra.

#### 2.2.2. Savitzky–Golay (SG) Filtering

Introduced by Savitzky and Golay, this technique facilitates the numerical differentiation of a dataset while incorporating a smoothing step based on a moving-window scheme [19]. The value at the central point is calculated via polynomial fitting over points covered by the window. Important parameters are the number of points (or window size) for the polynomial fitting and the degree of the polynomial itself. If the derivative is needed, the derivative of the given order is calculated on the results of the polynomial fitting. The maximum derivative achievable is bounded by the degree of the polynomial fitting; for instance, a third-order polynomial would allow estimation of up to the third-order derivative.

#### 2.2.3. Other Filtering Methods

Mean or median filtering in the spectral domain is a technique to smooth the data by replacing each point with the average or median of neighboring data points, respectively. The neighborhood is again defined by a window, where the width of the window is a free parameter of the method.

#### 2.2.4. Fourier Self-Deconvolution

Fourier self-deconvolution (FSD) is an advanced processing technique in infrared (IR) spectroscopy used to enhance the clarity of spectra with overlapping components [20]. Rooted in the principles of Fourier transformation, which decomposes complex signals into basic frequency components, FSD sharpens overlapping peaks, making them more distinct. In IR spectroscopy, where molecules absorb light based on their bonds and vibrations, overlapping peaks can occur, complicating the identification of individual molecules. FSD addresses this by enhancing the resolution, narrowing the peaks, and ensuring clearer separation. This method does not introduce new information but refines the existing data, making it crucial for analysts seeking precision in peak identification.

### 2.3. Baseline Correction

A measured IR spectrum is often distorted by a baseline contribution, which results from numerous instrumental and scattering effects. Baseline removal solutions may be as basic as subtracting an offset (DC shift) or deleting a piecewise created baseline by picking a few locations and connecting them with a linear function [21]. Polynomial fitting and differentiation based on SG filters are common baseline removal methods [19] (see above). Lower-order polynomials (second or third order) are often well suited for IR spectroscopy. Another approach is multiplicative scatter correction (MSC), which completes background correction and normalizing at the same time. MSC is a leading pre-processing method for NIR, and it was first introduced by Martens et al. in 1983 [22] and later refined by Geladi et al. in 1985 [23]. The main idea of MSC is to remove errors or imperfections, such as scatter effects, from the data matrix before modeling. MSC involves determining the correction coefficients, encompassing both additive and multiplicative components and subsequently adjusting the observed spectrum. In MSC, two main components are used. The first is $x_{org}$, the direct measurements determined via the (N)IR instrument. The second is $x_{ref}$, which is a matrix of reference spectra. We use $x_{ref}$ to help clean up and adjust the data from $x_{org}$ to make it more accurate. Basically, $x_{ref}$ helps us spot and correct any issues or errors in $x_{org}$. In many situations, the reference spectrum is the calibration set’s average spectrum. That said, it is also possible to use a generic reference spectrum. This approach was highlighted in the original work by Martens et al. [22]. The foundational version of MSC has evolved over time, leading to more advanced versions often referred to as EMSC, as documented in various studies [24,25]. This expanded version involves fitting the reference spectrum with a second-order polynomial, adjusting a baseline on the wavelength axis, and incorporating prior knowledge derived from the relevant spectra or potential spectral interferences. Therefore, methods like MSC and extended MSC (EMSC) are often used in IR spectroscopy. EMSC [21,26] handles baseline removal in addition to normalization, whereas MSC tries to normalize IR spectra. 

### 2.4. Normalization

Normalization is an essential step in vibrational spectroscopy. This is because the same material can have different spectra if recorded at different times or under different equipment settings, such as alignment or laser power. Even when the material is the same, the intensity can vary. Normalization aims to adjust these intensity differences, ensuring that the spectra of the same material recorded under slightly different conditions look as similar as possible. Normalization helps in tasks that measure spectral distances, improving the accuracy and working of the models, as shown in various studies [18,27,28,29]. This process adjusts for multiple things at once, such as changes in the equipment’s power, differences in how the sample scatters light or variations in sample thickness. There are many ways to perform normalization, as mentioned in many research papers [27,29]. The best method depends on the specific situation and the problem being addressed. Common methods include min–max normalization, the $l_{1}$ norm, the $l_{2}$ norm, and SNV normalization [30]. SNV adeptly addresses scatter effects and light absorption inconsistencies in spectral data. In the SNV method, each spectrum is first centered by subtracting the mean spectrum value from every data point. Scaling is then applied where every centered data point is divided by the spectrum’s standard deviation. Upon completing these adjustments, spectra exhibit a zero mean and a unitary standard deviation. Each of these methods has its own way of adjusting the data. Because most normalization approaches are applied to each spectrum separately, they are classified as one-way procedures [31]. 

The above pre-processing steps can be performed in different orders depending on the specific application and the characteristics of the IR spectra. It is important to note that selecting appropriate pre-processing methods can significantly impact the quality and accuracy of the results of the subsequent data modeling. Therefore, careful evaluation and optimization of the pre-processing steps are crucial for reliable and meaningful analysis in IR spectroscopy. After the pre-processing, artificial intelligence methods are required to extract information from the IR spectra. The objectives of an analysis task can be classification, regression, clustering, or dimension reduction. Selecting and constructing an optimal artificial intelligence method is task and data-oriented. In the following section, we will briefly overview the existing approaches that are most commonly employed in IR spectroscopy-based biological applications.

## 3. Data Modelling and Data Analysis

In the field of IR spectroscopy, data modeling and analysis play a crucial role in extracting meaningful information from complex spectral datasets. Therein, the trends, patterns, and relationships between spectral variables and sample properties are to be identified. Existent data modeling methods are normally classified into unsupervised and supervised methods. Unsupervised approaches are used to visualize the dataset or when independent variables like class labels are unavailable. Approaches in this category are divided into two groups: clustering and feature extraction, with examples of hierarchical cluster analysis (HCA) [32] and k-means clustering (KMC) [33] for clustering and principal component analysis (PCA) [34], successive projections algorithm (SPA) [35], uninformative variable elimination (UVE) [36], and autoencoder (AE) [37] for feature extraction. Supervised methods can be applied with access to the annotated dataset, which means that the independent variable to be predicted is given. The algorithm learns a model to relate the data to the required output, e.g., class labels or concentration of certain substances. After being well validated, the parameters of a constructed model are stored for future use to predict the new data. Supervised methods can be categorized into classification and regression. The separation between classification and regression, however, is not strict, as classification is achieved in most cases via regression models.

A support vector machine (SVM) [38], linear discriminant analysis (LDA) [39], random forest (RF) [40], k-nearest neighbors (KNN) [41], and artificial neural networks (ANN) [42] are among the classification methods that have been widely used for IR spectra. Commonly applied regression methods are multiple linear regression (MLR) [43], principal component regression (PCR), and multivariate adaptive regression splines (MARS) [44]. An overview of analysis models commonly used in IR is presented in Figure 4.

### 3.1. Classification

Classification in the field of IR spectroscopy aims to group spectra into different classes based on their distinct patterns or features. Typically, a model is trained on known IR spectra and used to predict new, unknown spectra. In this section, we will explain the methods that are commonly used for IR data, such as support vector machine (SVM) [38], linear discriminant analysis (LDA) [39], random forests (RF) [40], k-nearest neighbor classifier (KNN) [41], and artificial neural networks (ANN) [42].

#### 3.1.1. Support Vector Machine (SVM)

SVMs are a set of supervised learning algorithms broadly used for classification (and regression). The main idea is to search for a hyperplane that can separate two groups with the largest margins between the nearest data points from different groups. These closest data points, termed support vectors, play a vital role in determining the position and orientation of this hyperplane. A standout feature of SVM is its capability to manage non-linearly separable data through the kernel trick. This method involves transforming the data to a higher dimension, rendering it linearly separable. Common kernel functions include linear, polynomial, radial basis function, and sigmoid. Since its first introduction to chemometrics in reference [38], SVMs have been effectively used for mid and near-infrared spectroscopy, such as material identification [45,46], food discrimination [47], and medicine [48]. For instance, SVM modeling has effectively differentiated brain cancer and non-cancer spectra. Another significant study involved analyzing serum using ATR-FTIR and SVM [49]. In this study, nine different readings were taken from each of the 724 patient’s blood samples. The SVM model was trained using these data. Later, 104 more patients were added to check the model’s effectiveness. The model was able to identify and separate cases and non-cases pretty well. Devos et al. [50] looked at how hyper-parameters, like regularization and kernel width, affected the performance of their model. The results obtained, along with the data arrangement, helped to understand which samples were most important for their study. This also helped them pick out samples that were representative or typical for their research. They tested their ideas using two different NIR datasets: one with two groups and another with three groups. Hands et al. [51] used ATR-FTIR spectral data and an SVM approach to differentiate cancerous from non-cancerous gliomas. They trained the SVM with two-thirds of the patient data and tested it with the rest. The results showed that for the whole dataset, the SVM was quite accurate in identifying the disease from the control group. However, it is worth noting that SVM is not always the best choice for every dataset. This is especially true for datasets with more variables (features) than the number of samples (instances). SVMs could be hindered by a curse of dimensionality. In such cases, the risk of overfitting rises, leading to poor generalization performance. 

#### 3.1.2. Linear Discriminant Analysis (LDA)

LDA is a classification approach that seeks a linear subspace that maximizes the separation between the groups while minimizing the variance within each group. It is typically combined with partial least squares regression for spectroscopic analysis, specifically the method known as PLS-DA [39,52,53]. PLS-DA produces latent variables (LV) that transform spectral data into a new feature space, enhancing the differences between classes. Thereafter, LDA is performed in the new space to achieve classification. In reference [54], PLS-DA was employed alongside high-throughput FTIR spectra for disease diagnosis in human plasma. Using near-infrared diffuse transmission spectra, the researchers set out to distinguish between three distinct blood species: macaque, human, and mouse. According to both blind and external tests, the PLS-DA model exhibited exemplary performance in accurately distinguishing the three blood types. In a different study [55], researchers analyzed ATR-FTIR spectra from blood serum to diagnose children and adolescents with autism spectrum disorder. The study highlighted the impressive ability of PLS-DA to make accurate predictions. This high accuracy in predictions was linked to noticeable differences in the protein regions of the ATR-FTIR spectra. Unfortunately, while the approach performs well in two-class situations, it struggles and produces poor results when more classes are involved [56]. 

#### 3.1.3. Random Forest (RF)

RF is a promising decision tree-based classifier (or regression method) that has shown potential in IR processing data from different bio-medical samples like biofluids [57]. Essentially, a cascade of binary decisions is created based on subsets of variables, as illustrated in Figure 5 [40]. These decision trees are trained by iteratively splitting data to discover the optimal method for grouping it, which aims to reduce the Gini impurity [40]. A node’s Gini impurity is calculated according to Equation (1), where $g\left(\tau \right)\;$ is the node’s impurity, n is the total number of spectra at the node, and $n_{A}$ and $n_{B}\;$ are the numbers of spectra belonging to class A or B, respectively (cancer or non-cancer). The process is repeated on subsets of data characteristics until a Gini index of 0 is achieved. A Gini index of 0 indicates that the data within a node are perfectly pure, meaning all samples in that node belong to a single category or class.
(1)$$g\left(\tau \right)=1-{\left(\frac{n_{A}}{n}\right)}^{2}-{\left(\frac{n_{B}}{n}\right)}^{2}$$

RF has emerged as a prominent tool in IR spectroscopy-based applications. For instance, a study in reference [58] successfully employed RF in conjunction with NIR spectroscopy to ascertain the indigotin content in cream. RF classifiers were shown to be more robust, less susceptible to overfitting, and had a lower demand in parameter tuning, making them relatively more user-friendly than several other classifiers. RF models have proven successful in various research fields and are recognized for their ability to manage enormous datasets. Among those, RF with 500 distinct trees was used to identify cancer and non-cancer in serum samples based on ATR-FTIR spectroscopy [59]. In work [60], a combination of Fourier-transform mid-infrared spectroscopy and near-infrared sensor technologies were utilized alongside single spectra analysis and a multi-sensor information fusion strategy. The RF model was constructed to facilitate the origin identification of *Panax notoginseng*. The RF identification model demonstrated a high level of accuracy, at 95.6%. The RF analysis was additionally applied in brain cancer diagnosis based on IR spectroscopy of digitally dried serum samples [61]. One of the most significant features of RF is its adaptability, as it can be used for both regression and classification tasks and identify the most important and responsible features for classification [62]. While they might surpass SVM and PLS-DA in several metrics, they are computationally intensive. Predictions might become time-consuming, especially when the forest encompasses many trees [63].

#### 3.1.4. K-Nearest Neighbor (KNN)

The k-nearest neighbor method is another supervised classification technique that has proven beneficial for the classification of IR spectra. The method categorizes new data to the majority of its k-neighboring data points. Lechowicz et al. used ATR-FTIR serum data and KNN to distinguish between patients with rheumatoid arthritis and healthy controls [64]. Another study by Gajjar et al. employed KNN with various feature selection techniques on ATR-FTIR spectra to identify ovarian and endometrial cancers. Their results showed that the KNN method could identify ovarian cancer from serum samples and spot endometrial cancer from plasma samples [14]. In another study [41], researchers applied KNN and LDA methods to classify purple sweet potatoes, white sweet potatoes, and their adulterated samples. Despite its efficacy in many scenarios, KNN does have its limits. The method can be computationally burdensome when faced with large datasets, especially those with numerous dimensions. To mitigate this, it is crucial to adopt feature-extraction techniques before applying KNN to spectroscopic data. In this perspective, as Gajjar et al. highlighted, the choice of feature-selection methods can significantly impact the performance of KNN. This underscores the importance of carefully selecting and evaluating the feature extraction and selection methods for specific datasets [14].

#### 3.1.5. Artificial Neural Network (ANN)

Advancements in artificial intelligence technology have facilitated the development and use of artificial neural networks in disease diagnosis investigations [65,66]. ANNs are systems that imitate the functioning of the human brain and are widely used in various analyses. Fundamentally, ANNs are comprised of layers, including an input layer, an output layer, and variable numbers of hidden layers. They are commonly used to map inputs to outputs, depending on the type of analysis required for a specific dataset [67]. A prototypical representation of an ANN can be visualized in Figure 6. The output of each node is the weighted sum over all its input nodes plus a bias B. Additionally, activation functions are used to select which neurons are activated and passed on to the next hidden layer. The final classification is determined by the class with the highest probability (output). The ANN classifier is trained in a supervised manner through multiple iterations based on back-propagation algorithms. In each iteration, the error between true labels and the prediction is calculated, which is then back-propagated through the layers to adjust the weights of each layer and improve the output. Several papers have shown the promising performance of ANNs for classification tasks [68,69]. With the help of back-propagation, the ANN model is adjusted to map input to output with minimal errors and is, in principle, more flexible and better than other models. In theory, the more data supplied to a neural network, the greater its predictive potential; however, neural networks sometimes require a huge volume of input data in order to achieve high prediction accuracy [66]. Furthermore, it can need a lot of computer power, and training can take a long time—days or even weeks if the model is too complex or has a large dataset.

### 3.2. Regression

Univariate linear regression (LR) is a statistical technique used to model the relation between a dependent variable Y and an independent variable X and is expressed through Equation (2).
(2)$$Y=\beta_{0}+\beta_{1}X+\varepsilon$$
where $\beta_{0}$ represents the offset, $\beta_{1}$ denotes the slope of the regression line, and ε is the residual term, which reflects the deviation between the observed and forecasted values. However, univariate linear regression, which contains hundreds of variables, is rarely utilized in vibrational spectroscopy. One study [70] proposed a novel method for quantitatively determining the Si/Al ratio (SAR) in zeolite frameworks using IR spectroscopy. The method analyzes the intensity of the IR band at specific wavenumbers and correlates this intensity to the SAR with univariate linear regression. In another study, the seed content in red pepper powders was derived using FT-IR spectroscopy with canonical discriminant and multiple linear regression analyses (MLR) [43]. Researchers studied 165 red pepper powder samples with different seed amounts using spectroscopy. They used 21 main data points to classify the samples correctly based on seed content, achieving a high success rate. They also used a method called MLR to figure out the seed content of new samples without needing complex statistics. Plus, they looked into using a combination method called principle component regression, which mixes PCR and LR to obtain even better prediction results. In study [71], scientists compared the effectiveness of two biased regression models for predicting diabetic and normal serum using FTIR spectroscopy data. The study worked with data from 11 normal samples and utilized regression for diagnostic classification. The results showed that both partial least squares regression and principal component regression produced similar outcomes. However, they emphasized the need for more research to determine which method might be superior. In study [72], researchers explored various regression methods to link soil spectra with soil organic carbon (SOC) and total nitrogen (TN). They specifically used multiple linear regression to associate specific reflectance from the NIR region with SOC and TN levels. In another study [44], the team employed multivariate adaptive regression splines (MARS) to estimate SOC and other related soil properties. 

Overall, in the field of disease diagnosis, supervised learning has been found to produce high accuracy. This has helped researchers investigate the differences between the disease states and thereafter improve disease categorization. However, supervised procedures are not always appropriate, especially if no confident group annotation is available, as it is during exploratory research. In these cases, unsupervised techniques can be more suitable. An unsupervised approach can provide useful insights for classification and can be used as a preliminary step before applying supervised methods as a feature extraction tool. When dealing with huge datasets that may not be suitable for ANN, unsupervised approaches such as clustering can be more appropriate. Clustering is a popular unsupervised method that can identify structures or patterns in a dataset, allowing data classes to be identified [33,73]. 

### 3.3. Clustering

Clustering aims to group data such that the data points within a cluster are more alike to one another than those in other clusters. K-means clustering (KMC) and hierarchical clustering are the most frequently used clustering algorithms in IR spectroscopy. However, when extra information is given to the algorithm, such as the k value, cluster sizes, and labeling of data subsets, it is referred to as a semi-supervised approach [74,75]. KMC clusters data points into k clusters, where k is the user-specified number of clusters [33,73]. The algorithm starts by randomly selecting and assigning the spectra to the k cluster as the initial cluster centers. Each data point is then assigned to the cluster it is closest to, based on its distance (mostly Euclidean distance) to the cluster centers. Once all data points have been assigned to a cluster, the center of each cluster is updated. This process is repeated until the variance inside each cluster is minimized. The algorithm is then run multiple times with different initial cluster centers to ensure that the result is not biased by the initial choice of cluster centers, e.g., the distribution. The best result across all runs is chosen as the final clustering. KMC is widely used in various fields, including image segmentation, text mining, and bioinformatics, for grouping similar data points together and discovering patterns in large datasets. HCA is another popular unsupervised machine learning algorithm used for clustering. Unlike KMC, the number of clusters does not need to be predefined in HCA. HCA generates a cluster hierarchy, also known as a dendrogram, that shows the hierarchy of clusters and how they are related to each other [32]. There are two main methods for hierarchical clustering: top-down (also called divisive) and bottom-up (also called agglomerative) clustering. In the bottom-up approach, each data point starts in its own cluster, and then pairs of clusters are successively merged based on their similarity until all data points are in one cluster. In contrast, the top-down approach starts with all data points in a single cluster and recursively divides it into smaller sub-clusters until each data point is in its own cluster. The optimal number of clusters for the dataset may be determined using the dendrogram. Similar to KMC, the Euclidean distance is commonly used to measure the similarity between clusters in HCA. However, various other distance metrics can be used for several types of data. For example, cosine similarity is commonly used for text data, and correlation-based distances are used for gene expression data, as various approaches have been described in references [76,77]. 

In recent years, both KMC and HCA have been successfully used to identify biofluid classes using IR spectroscopy [74,75]. Backhaus et al. [42] studied how well KMC analysis worked using two different methods of FTIR: transflection and transmission. They tested it on samples from 193 breast cancer patients and achieved promising results for the transmission method and even better results for the transflection method, which were similar to the results of using artificial neural networks. In a different research, scientists used hierarchical cluster analysis on ATR-FTIR spectra from bladder wash samples taken during a procedure called cystoscopy. They found that the ability to detect bladder cancer was consistent, no matter which part of the spectra the analysis was performed, with both sensitivity and specificity being around 81% [78]. One of the advantages of using KMC as a classification technique is that it is straightforward and easy to apply, requiring only one input parameter, k. Additionally, KMC offers several parameters that may be adjusted to improve the algorithm’s performance. However, the cluster centres in KMC are highly dependent on the initial cluster centre selection, which is determined using the Euclidean distance [79]. Both KMC and HCA are prone to noisy data and outliers, which can significantly impact their prediction. Modifications to the algorithm can be made to address this issue, but this requires a deeper understanding of the approach. HCA generates a visual map, known as a dendrogram, indicating how the data may be segregated without requiring prior knowledge of the number of groups. However, it has several limitations, including its computational cost, making it unsuitable for large datasets [80]. Before applying HCA, the dataset may be reduced using principal component analysis to minimize dimensionality [78]. Furthermore, a clustering method cannot predict new data, making it a poor diagnostics tool.

### 3.4. Feature Extraction

In the realm of multivariate data analysis, a particular dataset may consist of variables that exhibit strong correlations. In such cases, the ability to describe the dataset concisely using fewer variables while capturing underlying correlation patterns can be extremely beneficial. The original variables can be linearly combined to generate new variables to achieve this. This process is known as dimensionality reduction and involves the generation of new data space with a reduced dimension. 

Dimensionality reduction also helps in extracting important signals from irrelevant noise, thereby improving the accuracy of classification and clustering techniques. In addition, the computational complexity of many classifications and clustering techniques is high. Before supplying the data to these classification algorithms, it makes sense to convert the original variables to a lower-dimensional space. One such frequently used dimensionality reduction method is principal component analysis (PCA). 

PCA is one of the unsupervised methods used in the bioinformatics and chemometrics field. The goal of PCA is to optimize the variance within the whole dataset, and it requires no prior knowledge of the data. PCA reduces data dimensionality by geometrically projecting the dataset onto fewer dimensions, referred to as principal components (PCs), establishing a new coordinate system in which the first principal component (PC1) describes the direction of the largest variance in the dataset, the second principal component (PC2) describes the second largest variance, and so on [34,81]. 

Scores and loadings are the two components of the transferred data. Scores represent the value of a sample when projected orthogonally onto the PCs, while loadings indicate which variables are most closely related to each PC, highlighting sources of variance within the dataset. Although there are other ways to reduce dimensionality, PCA is by far the most used in spectroscopy [82], especially when the datasets are vast, and the spectral markers used to distinguish between classes are unclear. 

When the data were processed using PCA, an FTIR investigation of plasma samples from patients with various stages of prostate cancer and controls revealed a strong distinction between the two classes [83]. The data from the bladder wash research were also analyzed using PCA, and the findings successfully distinguished bladder cancer patients from controls [78]. The use of PCA has proven to be beneficial for data collected through two distinct measurement modalities: transmission FTIR (with KBr pellets) and ATR-FTIR (with dried wash samples). The PC1 vs. PC2 scores exhibited significant discrimination between cancer patients and control samples, indicating the potential for capturing underlying correlation patterns within the data set. Subsequently, the PC loadings were analyzed to identify the spectral characteristics that contributed the most to the observed variation. These spectral characteristics were then subjected to cluster analysis for further investigation. 

When combined with supervised classification algorithms, PCA serves as a powerful feature extraction tool [84]. By reducing the dimensionality of the data, PCA enables faster class discrimination based on the actual spectral differences within the dataset instead of relying on pre-assigned spectral differences across classes [85]. Another significant feature of PCA is its capacity to handle noisy datasets; by lowering the dimensionality of the dataset, noise is removed. 

It is important to note that PCA is sensitive to the scale of the data; thus, data normalization is crucial, particularly if exploring data other than spectroscopic data (for example, protein levels). Categorical characteristics must also be transformed into numerical values prior to applying PCA. While PCA is not a sufficient classifier on its own, it is frequently utilized as a feature selection method prior to classification analysis [85,86].

The successive projections algorithm (SPA) is a variable selection method that is gaining considerable attention in analytical chemistry. This article [35] introduces the fundamental characteristics of SPA as applied to multiple linear regression and linear discriminant analysis. It also discusses various adaptations of SPA that have been proposed for effective sample selection. In reference [36], a combination of uninformative variable elimination (UVE) and SPA is proposed as an approach for variable selection in multivariate calibration. In this strategy, UVE is employed to identify informative variables, while SPA is subsequently used to select variables with the least redundant information from the previously selected informative variables. This proposed method was effectively utilized in the analysis of nicotine in tobacco lamina using near-infrared (NIR) reflectance data. 

As a new method of dimension reduction, a deep autoencoder represents powerful capability in feature extraction with both linear and non-linear relationships [37]. Autoencoders consist of an encoder that maps the input data to a latent representation and a decoder that reconstructs the input data from the latent representation. In reference [87], an AE has been evaluated as a feature extraction technique for near-infrared spectroscopy-based discriminating analysis. The study used four inputs for sample discrimination: AE-extracted features, raw NIR spectra, principal component scores, and locally linear embedding features. The results showed that using AE-extracted features significantly improved the discrimination accuracy of all eight products, particularly when the sample spectral features were indistinct. In another study, a novel approach that utilizes a one-dimensional convolutional autoencoder in conjunction with near-infrared spectroscopy was introduced for analyzing the protein, moisture, oil, and starch content of corn kernels. The study involved the creation of a one-dimensional convolutional autoencoder model for three different spectra in the corn dataset, which yielded thirty-two latent variables per spectrum, representing a low-dimensional representation of the spectrum. The study then developed multiple linear regression models for each target using the latent variables obtained from the autoencoder models [88]. 

### 3.5. Evaluation Metrics for Classification and Regression Models

In the realm of machine learning, assessing the performance of models is crucial to ascertain their performance and applicability. The true power of classification is realized when it can accurately categorize a given sample into its correct category or class. Once a classification model is developed, it is imperative to verify its performance. This ensures the model is robust and can reliably predict the correct category for new or unknown samples. To confirm a model’s performance, several metrics are utilized, including sensitivity, specificity, accuracy, precision, and F1 score. Sensitivity (also known as the true positive rate) measures the proportion of actual positives that are correctly identified. In other words, it quantifies how many of the samples that truly belong to a particular class are correctly classified by the model. In contrast, specificity (or the true negative rate) measures the proportion of actual negatives that are correctly identified. It indicates how many of the samples that do not belong to a particular class are correctly identified as such by the model. Accuracy calculates the proportion of correct predictions out of all instances. Precision assesses the accuracy of positive predictions, and the F1 score harmonizes precision and recall. Evaluating regression models involves a suite of metrics that assess prediction accuracy and the model’s explanatory power. The mean absolute error (MAE) and mean squared error (MSE) measure average prediction errors, with MSE emphasizing larger errors by squaring them. The root mean squared error (RMSE) further refines MSE by bringing it to the original unit scale. R-squared gives insight into how well independent variables explain variations in the dependent variable, and its adjusted counterpart considers the number of predictors in the model. MAPE focuses on percentage-based prediction errors, and residual plots visually convey the distribution of these errors, illuminating model fit nuances. Each metric provides a unique perspective on the evaluation of classifiers and regression models, aiding in identifying areas for improvement.

## 4. Deep Learning in the Analysis and Pre-Processing of IR Spectroscopy

IR spectroscopy provides detailed insights into the molecular characteristics of a sample by identifying its specific vibrational frequencies. The vast amount of data produced in each spectral analysis is a rich source of information but also presents challenges in extracting relevant insights. While traditional pre-processing holds pivotal importance in chemometrics, recent evidence suggests that its significance diminishes with the expansion of sample size [89]. As a result, when dealing with large datasets, methods such as PLS may fail in capturing intricate mixed patterns inherent to data, including essential absorption and scattering features [90]. This limitation underscores the need for methodologies that can adeptly recognize non-linear patterns. With its robustness in handling large data, deep learning has proven high potential in this context [91,92,93]. The growing interest in DL stems from its ability to continually learn as more data become available, whereas conventional machine learning approaches often see their performance level off after a certain threshold. However, when it comes to implementing deep learning for spectral data, the standard 2D convolutional neural networks (2D-CNN) found in computer vision tasks are not directly applicable. Instead, the unique structure of spectral data necessitates using one-dimensional convolutional neural networks (1D-CNN). This requirement arises from representing a single spectrum as a 1×n dimension, where n denotes the number of wavelengths measured [93]. Deep learning techniques may be used in a variety of studies, including data cleaning, classification, clustering, and other processes, so we will concentrate on them in this section. 

### 4.1. Deep Learning in Pre-Processing

Interpretation of infrared spectroscopy can be challenging because of distortions caused by scattering, equipment-related issues, and various physical irregularities. Traditional pre-processing steps, while effective to some extent, may not comprehensively account for the inherent complexities and minute nuances present in IR spectra. Deep learning, particularly convolutional neural networks (CNNs), offers unique advantages for IR spectroscopy pre-processing. Deep learning models, by design, can automatically learn and extract crucial features without explicit manual intervention. This adaptive learning provides robustness against a broader range of spectral distortions. Compared to standard pre-processing techniques, CNNs are less reliant on prior knowledge, can handle a wider variety of distortions simultaneously, and improve spectral analysis’s overall efficiency and accuracy. The depth and intricacy of these neural networks enable them to discern patterns and correct anomalies that might be overlooked or inadequately addressed via conventional methods. Adopting deep learning in IR pre-processing promises enhanced clarity, precision, and reliability in spectral interpretation [94]. Recently, a stacked auto-encoder-based network was used to illustrate the power of deep learning in Mie scattering correction [95]. The network was demonstrated to learn the steps involved in the RMieS correction by substituting a regression layer for the last layer of the pre-trained network. The authors [96] used a one-dimensional U-shape convolutional neural network to remove the artifacts caused merely by optical influences. A simulated dataset made up of apparent absorbance and absorbance pairs was produced for PMMA spheres in accordance with the Mie theory to do this. Using this dataset, a 1D U-Net was trained to convert the input apparent absorbance into the corrected absorbance following a process of data augmentation. A six-layer CNN model, as mentioned in reference [97], was developed for the quality identification of three types of macadamia nuts. This model achieved a perfect accuracy of 100%. This innovation offers dependable technical backing for an efficient and cost-effective quality assessment of macadamia nuts.

### 4.2. Deep Learning for Data Modeling

Besides pre-processing, deep learning also leverages extract patterns and makes predictions or classifications. Given its ability to process vast amounts of data and automatically extract features, deep learning models, especially CNNs, have shown tremendous promise in a multitude of applications. As stated in reference [98], a one-dimensional convolutional neural network was introduced for distinguishing aristolochic acids and their analogs in traditional Chinese medicine. The comparative analysis revealed that traditional shallow learning algorithms heavily depend on feature extraction algorithms like PCA or t-SNE. Meanwhile, the 1D-CNN model demonstrated remarkable accuracy, even without using any feature extraction algorithms. Reference [99] outlines the application of 1D-CNN, 2D-CNN, and PLS-DA for determining the geographical origin of tobacco leaves. According to the experimental results, both the 1D-CNN and 2D-CNN demonstrated superior performance compared to PLS-DA. Infrared spectroscopy gathered serum samples from patients with thyroid issues and healthy individuals. To expand the data, these samples were mixed with different noise levels. The enhanced data were then processed by three models: MLP, LSTM, and CNN. The effectiveness of the models was tested using a 10-fold cross-validation method. Once the data were evaluated, the accuracy rate for CNN was bigger than the other models [100]. Researchers in reference [101] introduced a new diagnostic method that combines FTIR and Raman spectroscopy, benefiting from their detailed molecular information. They took spectra from the serum samples of healthy individuals and patients with various cancers. The spectral data were combined in different ways and then processed using three models: SVM, CNN-LSTM, and MFCNN. The results showed that the MFCNN model, designed especially for these kinds of data, performed better than the traditional models. The authors of reference [102] used FTIR spectra from yeasts, fungi, and bacteria to test how well they could identify them at the genus and species levels. They compared how well a random forest method worked on adjusted spectra with how deep learning performed on expanded spectra. Adjustments and expansions were made for both technical and biological repeats. Furthermore, in the realm of NIR, the stacked autoencoder (SAE) and variational autoencoder (VAE) are two frequently utilized deep architectures. An SAE is composed of an encoder that captures a condensed representation of the input data and a corresponding decoder that attempts to recreate the original input. Notably, the decoder mirrors the encoder in terms of layer count and hidden neuron configurations. Given its multilayer nature, an SAE tends to settle into a local minimum. A dual-step learning methodology is adopted to optimize its convergence. Initially, a single-layer autoencoder (AE) deduces the weights for the encoder–decoder pairs layer-by-layer in a procedure termed greedy layer-wise learning. This is followed by refining the entire SAE. Thus, the SAE learning process bifurcates into a single-layer AE phase and a subsequent multilayer AE refinement stage. Highlighting its practical application, Wojciech and his team from AGH University [103] employed three distinct models to assess soil and terrain: an SAE, a CNN, and a composite model. This composite model amalgamates various multilayer perceptron algorithms, introducing two separate regression estimation techniques to interpret the Vis-NIR spectral signals [103]. Fu and his team created a model that mixes a stacked sparse autoencoder with a special support vector machine. They made this to study near-infrared data from maize seeds. Their goal was to correctly identify different types of maize seeds [104]. Due to the limited training samples for NIR data, GAN is used to create more training samples [105]. A GAN has two parts, including a generator network, which makes data from a random vector based on a Gaussian distribution, and a discriminator, which checks if the data made via the generator look like the actual data. Different architectures like fully connected neural networks, CNN, or RNN can be chosen as the generator and discriminator depending on the kind of data. For NIR data, using a combined CNN + RNN setup for both the generator and discriminator is suitable because it can pick up on local features and also understand the long-term connections between wavelengths. Facing challenges like not having enough samples within a category and uneven distribution of samples between classes. Zheng and his team suggested a tweaked bidirectional generative adversarial network (BI-GAN) approach for classifying near-infrared spectra of drugs. The findings show that models based on CNN performed better than the stacked autoencoder, which had the highest error rates in estimating soil properties [106]. 

Overall, in this section, we highlighted the potential of deep learning as an artificial intelligence-based technique for modeling IR data in a more effective and accurate manner. Deep learning is recognized as a reliable and automated data science tool that can extract valuable information from complex data, thereby enhancing the usability of such data across different fields. In addition, while deep learning techniques can be utilized for various purposes, such as data cleaning, classification, clustering, and more, this section specifically focuses on their application in IR data analysis.

## 5. Feasibility of Machine Learning Approaches for High-Resolution Spectroscopy

With the innovations in measurement techniques, IR spectroscopy has become more and more attractive for high-resolution and/or high-throughput measurements to capture more detailed information about a sample within a shorter time. This leads to further requirements and demand for big data computation and analysis. Machine learning, especially deep learning approaches, has been playing an increasingly important role in this context, thanks to the advances in computational facilities like GPUs, server services, and public datasets. On the other hand, deep learning has been largely applied to elevate the resolution of a measurement, which has been shown to have a high potential to break the limit of the instrument at a lower cost. For example, a new high-resolution method was introduced by combining compressive sensing theory with deep learning. Their approach has two steps. First, they use the sparse nature of CS theory to create a high-resolution image with more detailed information. Then, they use a trained network to eliminate noise from the previous step and add more detail to the training data. This effective method lets them obtain clearer infrared pictures without needing expensive or complex infrared sensors [107]. In reference [108], researchers proposed a method called classified dictionary learning to make clearer, high-resolution infrared images. It sorts sample features into logical groups and develops a dictionary for each group. The best dictionary pair is then picked for each image enhancement. This method improves results without needing more computing power or taking more time.

## 6. Conclusions

Researchers have been exploring the integration of artificial intelligence-based approaches, such as chemometrics, machine learning, and deep learning, with spectroscopic measurements in chemistry and chemical data for several years. These data analysis methods have gained popularity and significant application prospects across various industries, from the food sector to medicinal applications. In this context, modern researchers are harnessing the capabilities of data science tools, aiming not only to automate information extraction from intricate data but also to unlock novel avenues for applying these data across diverse sectors. This review paper focuses on the latest research on AI-based methods for IR spectroscopic measurements. Specifically, machine learning and deep learning techniques, such as CNN, are used to process IR data, and data modeling for a range of tasks is reviewed following the enhancement of data quality. The studies highlighted in this review demonstrate how machine learning algorithms can be used to analyze IR spectra and how powerful they can be. In combination with machine learning, spectroscopy has the potential to significantly improve prediction and classification methods and, consequently, daily operations. To summarize, the primary thrust of this review is to navigate the intricacies of data pre-processing, modeling, and comprehensive analysis techniques specifically tailored for managing IR spectral datasets.

## Figures and Tables

**Figure 1 molecules-28-06886-f001:**
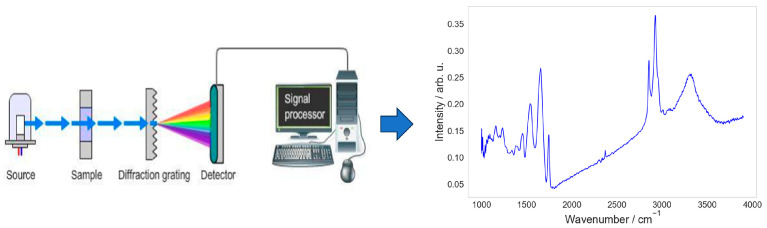
Relation between matter and electromagnetic radiation. After interacting with the material, infrared light is split into its separate frequency components via a monochromator, and a photodiode array detector can be used to determine which frequencies were absorbed [4].

**Figure 2 molecules-28-06886-f002:**
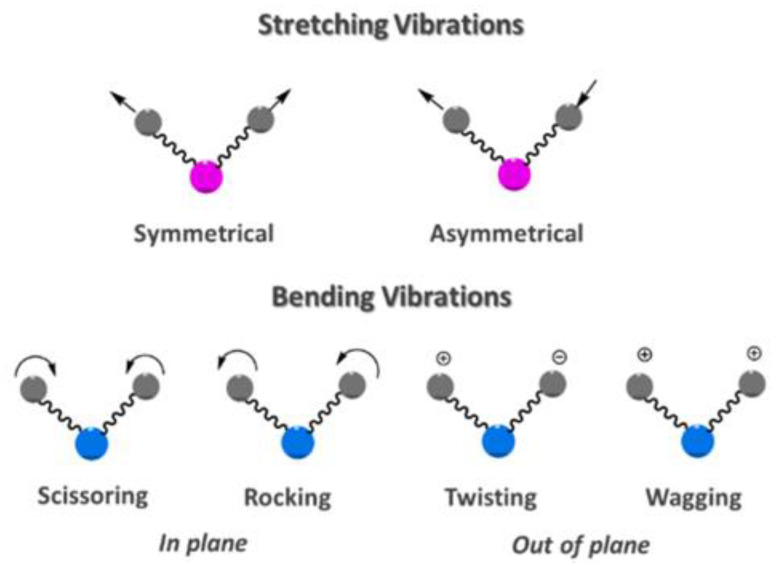
Different types of vibration (reprint from ref [5]). Stretching vibration and bending vibration are two types of vibrations that can be measured using IR spectroscopy. In the illustration, the circles in pink/blue symbolize the central atom around which the vibrations occur, while the gray circles represent the surrounding atoms that are bound to it.

**Figure 3 molecules-28-06886-f003:**
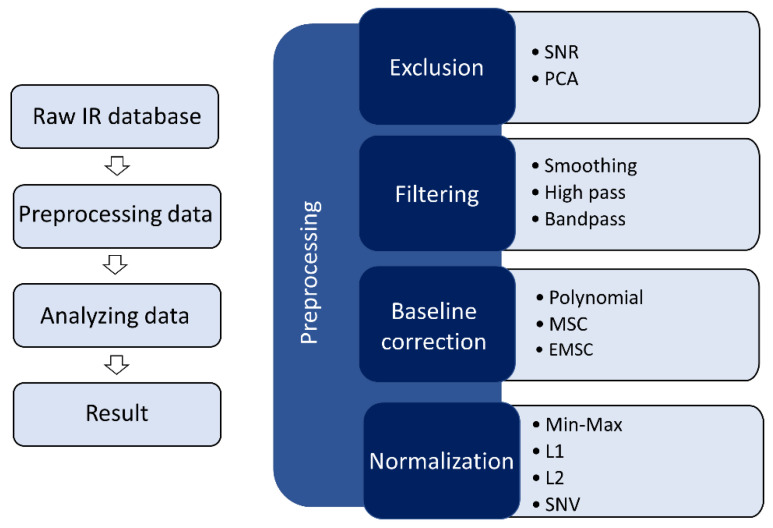
Schematic of the pre-processing pathway for IR spectra. Many pre-processing techniques are utilized when a whole dataset has been collected, including exclusion/cleaning, background correction, normalization, and smoothing. Then, the dataset is ready for further analysis.

**Figure 4 molecules-28-06886-f004:**
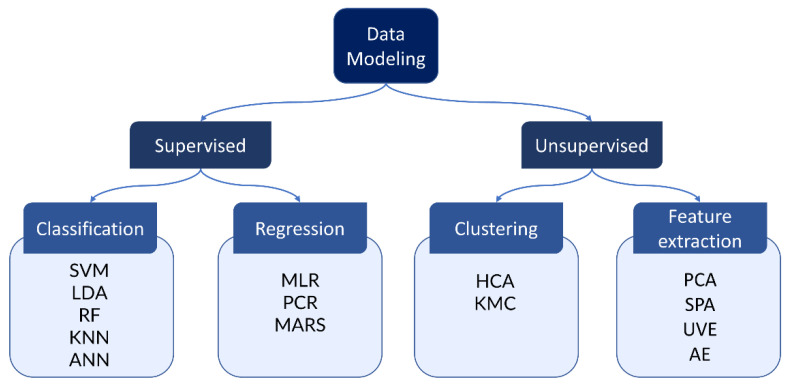
An overview of analysis models for IR spectra. Existent models can be categorized into supervised or unsupervised methods. Classification and regression are typically achieved via supervised methods, where the input data are annotated, and the algorithm learns to relate the spectral signal to the desired label output. In contrast, unsupervised methods such as clustering and feature extraction do not use the label information. Instead, these methods try to find the characteristic patterns or structures within the data.

**Figure 5 molecules-28-06886-f005:**
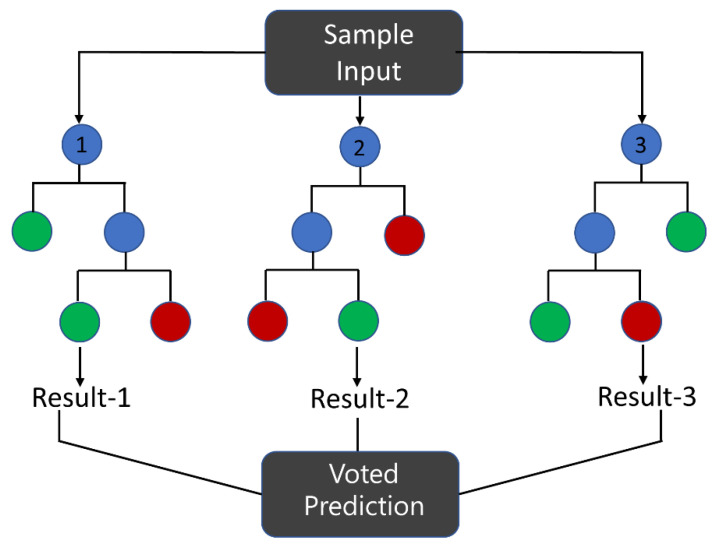
Visualization of a simple random forest (RF) with three decision trees. The blue circles symbolize the decision nodes, while the red and green circles denote the leaf nodes.

**Figure 6 molecules-28-06886-f006:**
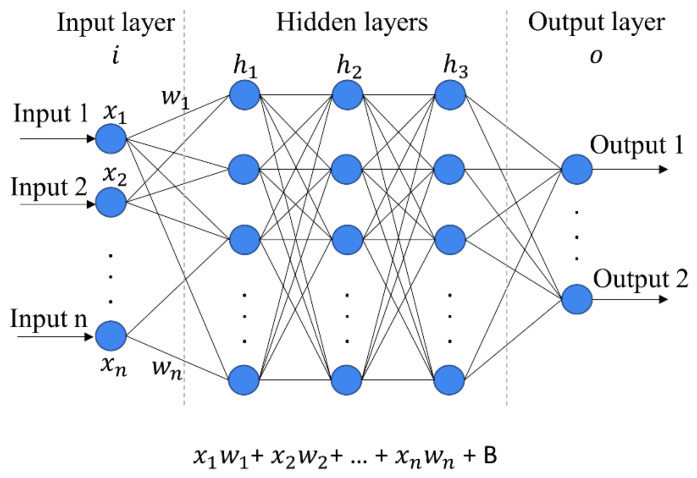
Representation of a feed-forward neural network (NN) with three layers: input, hidden, and output. The data flow from the input layer to the output layer via the hidden layer(s). Each artificial node is connected to every.

## Data Availability

Not applicable.

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
