# Peer review of "Exploring the Steps of Infrared (IR) Spectral Analysis: Pre-Processing, (Classical) Data Modelling, and Deep Learning"

_molecules, 2023, doi:10.3390/molecules28196886_

Round 1

Reviewer 1 Report

Review Report:

In their manuscript Mokari and Bocklitz demonstrated a pedagogical review of machine learning based data processing and how this technique can be employed for IR based data analysis. With this powerful approach, complex data analysis will become much faster, reliable and thorough.

This work is important in the context of the development of large data analysis. The manuscript is well-written, insightful, and thorough. Therefore, I recommend the publication of this manuscript essentially as is without any major modifications. My comments are copied below. I use the following abbreviation, P-page number.

Comments:

1. P2-Section 1: The authors are encouraged to mention the wavenumbers for different IR regions.

2. P2-Section 1: The authors should also mention the unit of IR-intensities (km/mol) before proceeding to normalized intensities.

3. I encourage the authors to discuss the feasibility of this machine-learning based approach for high-resolution spectroscopy.

Reviewer 2 Report

The basic problem with this manuscript is that it tries to cover too large area. The authors attempted to provide information both on principles and applications of various methods of data analysis. As a result, neither the principles nor the applications are described with the comprehension satisfactory for review paper. If the authors wants to keep the present form of this work, they should significantly enlarged its size. Alternatively, the may focus on a few selected methods in more comprehensive way. In the present form, this work is not suitable for publication.

No comments.

Round 2

Reviewer 2 Report

The authors did an effort and significantly modify the manuscript. In the present form it is suitable for publication.